# Transcriptome Analysis and Identification of Cadmium-Induced Oxidative Stress Response Genes in Different *Meretrix meretrix* Developmental Stages

**DOI:** 10.3390/ani14020352

**Published:** 2024-01-22

**Authors:** Yiyuan Xu, Chenghui Wu, Jianyu Jin, Wenhan Tang, Yuting Chen, Alan Kueichieh Chang, Xueping Ying

**Affiliations:** 1College of Life and Environmental Sciences, Wenzhou University, Wenzhou 325035, China; 21451335051@stu.wzu.edu.cn (Y.X.);; 2National and Local Joint Engineering Research Center of Ecological Treatment Technology for Urban Water Pollution, Wenzhou University, Wenzhou 325035, China

**Keywords:** cadmium, *Meretrix meretrix* larvae, transcriptome sequencing, oxidative stress

## Abstract

**Simple Summary:**

Cadmium (Cd) is a major aquatic pollutant that can cause toxic effects on aquatic animals, including the economically important clam *Meretrix meretrix*. A transcriptomic analysis was performed on *M. meretrix* fertilized eggs, D-shaped larvae, pediveligers, and postlarvae following exposure to Cd^2+^-containing seawater. A number of genes related to the oxidative stress response were found to be differentially expressed in the fertilized egg and the different larval forms when exposed to Cd^2+^. These included the *CCO*, *Ndh*, *HPX*, *A2M*, *STF*, and *pro-C3* genes, and changes in their expression levels in response to Cd^2+^ exposure were confirmed by *q*RT-PCR assays. In addition, *q*RT-PCR also revealed marked changes in the expression of two other oxidative stress-related genes, *Nrf2 and MT*, in the fertilized egg and different larval forms of *M. meretrix* in response to Cd^2+^ exposure. The results suggested that changes in oxidative stress-related genes might be crucial for *M. meretrix* to cope with the adverse effects of Cd during its development.

**Abstract:**

Cadmium (Cd) is one of the major pollutants in the aquatic environment, and it can easily accumulate in aquatic animals and result in toxic effects by changing the metabolism of the body, causing a serious impact on the immune system, reproductive system, and the development of offspring. The clam *Meretrix meretrix* is one of the commercially important species that is cultivated in large-scale aquaculture in China. To elucidate the underlying molecular mechanisms of Cd^2+^ in the developmental processes, fertilized eggs and larvae of *M. meretrix* at different developmental stages were exposed to Cd^2+^ (27.2 mg L^−1^ in natural seawater) or just natural seawater without Cd^2+^ (control), and high-throughput transcriptome sequencing and immunohistochemistry techniques were used to analyze the toxic effects of Cd on larvae at different early developmental stages. The results revealed 31,914 genes were differentially expressed in the different stages of *M. meretrix* development upon treatment with Cd^2+^. Ten of these genes were differentially expressed in all stages of development examined, but they comprised only six unigenes (*CCO*, *Ndh*, *HPX*, *A2M*, *STF*, and *pro-C3*), all of which were related to the oxidative stress response. Under Cd exposure, the expression levels of *CCO* and *Ndh* were significantly upregulated in D-shaped and pediveliger larvae, while *pro-C3* expression was significantly upregulated in the fertilized egg, D-shaped larva, and pediveliger. Moreover, *HPX*, *A2M*, and *STF* expression levels in the fertilized egg and pediveliger larvae were also significantly upregulated. In contrast, *CCO*, *Ndh*, *HPX*, *A2M*, *STF*, *and pro-C3* expression levels in the postlarva were all downregulated under Cd exposure. Besides the genes with changes in expression identified by the transcriptome, the expression of two other oxidative stress-related genes (*MT* and *Nfr2*) was also found to change significantly in the different developmental stages of *M. meretrix* upon Cd exposure, confirming their roles in combating oxidative stress. Overall, the findings of this study indicated that Cd would interfere with cellular respiration, ion transport, and immune response through inducing oxidative stress, and changes in the expression of oxidative stress-related genes might be an important step for *M. meretrix* to deal with the adverse effects of Cd at different stages of its development.

## 1. Introduction

Heavy metals are one of the aquatic environmental pollutants. Its toxicity and non-degradable characteristics make it the main target of environmental monitoring programs [1,2,3]. Cadmium (Cd) is one of the heavy metals found in the aquatic environment, and it has harmful effects on the survival and growth of aquatic organisms and their associated ecosystems [4,5]. After entering the aquatic environment, Cd can be accumulated by aquatic organisms and amplified in the food chain, leading to the gradual accumulation of the metal in aquatic organisms and eventually in humans [5,6,7]. Once inside the animals, Cd can induce significant oxidative stress by triggering the production of a large amount of reactive oxygen species (ROS), and these ROS can react with different biomolecules, such as DNA, proteins, and lipids, eventually leading to oxidative stress and cell death [7,8,9]. In order to cope with oxidative stress, aquatic organisms have independently evolved an antioxidant defense system capable of removing ROS and repairing damaged DNA [10,11], as well as sequestering metal ions by metal-binding proteins such as metallothionein [8]. 

Many studies on the responsive mechanisms of metal stress in mollusks have mainly focused on individual genes. Examples of such genes that have been commonly studied in mollusks are *Hps70*, *Hps90*, *MT*, *Nrf2*, *p38*, *p53*, *Bax*, *Blc2*, and *Caspase 3* [8,10,12,13,14]. Among these genes, the *Nrf2* (nuclear factor E2-related factor 2) and *MT* (metallothionein) genes play an important role in oxidative stress induced by metal exposure [15,16,17,18]. *Nrf2* is a key factor in the cellular antioxidant defense mechanism, and its expression level in medaka *Oryzias melastigma* larvae was shown to be down-regulated after exposure to 6 mg L^−1^ Cd^2+^ for 0.5 and 6 h [19]. *Nrf2* is a transcription factor that functions by initiating the expression of a series of antioxidant cytoprotective genes by binding to the antioxidant response element (ARE) within the regulatory region of these genes [15,20]. As for MT, it is a cysteine-rich, non-enzymatic antioxidant that can bind metals and participate in redox reactions [17,21]. The metal-binding ability of MT allows it to maintain a dynamic balance of metals in living organisms, slow down the production of free radicals, and participate in antioxidant defense reactions to protect cells from the toxic effects of heavy metals [22,23]. Therefore, the mRNA levels of *Nrf2* and *MT* can be regarded as an indicator of heavy metal pollution in aquatic organisms [24,25]. However, despite the numerous studies on a specific gene involved in the responsive mechanisms to metal stress, a more effective way to uncover a more extensive mechanism involved in metal tolerance in mollusks would be to look at the transcriptomic changes that occur under metal stress [12,26,27,28]. This would enable the changes in gene expression via changes in mRNA levels to be quantified, enabling the identification of key genes and their responsive gene modules under the impact of a specific environmental condition. Such an approach can greatly facilitate the analysis of the physiological functions that may change in response to the presence of such a specific environmental condition. 

Bivalves are easily exposed to metal pollution because of their benthic and filter-feeding lifestyles [2,29]. The life cycles of bivalves are marked by the formation of larvae, which appear in the early stage of development, a stage that is highly sensitive to heavy metals [22,30,31]. One of the important economic bivalves that is cultured on a large scale in China is the clam *Meretrix meretrix* [32]. This clam can accumulate heavy metals and other pollutants at high levels, and it has been gradually used as an indicator and model bivalve to study heavy metal pollution in a marine environment [1,33]. Among the toxic effects exerted by heavy metals on *M. meretrix* are growth inhibition [22], reproductive impairment [8,10], immunotoxicity, and oxidative damage [9,17]. However, the oxidative stress mechanisms associated with such toxic effects and the changes in gene expression patterns under metal stress in *M. meretrix* larvae at different developmental stages remain unexplored.

In this study, the RNA-seq approach was used to examine the changes in gene expression in the fertilized eggs, D-shaped larvae, pediveligers, and postlarvae of *M. meretrix* that occurred after exposure to Cd^2+^. A transcriptome sequencing library was successfully constructed using second-generation Illumina sequencing technology and used to screen for oxidative stress-related genes. The functions of these genes were identified, and their potential roles in the management of oxidative stress were discussed. The findings obtained from this investigation could provide further insight into the mechanisms of oxidative stress associated with *M. meretrix* larvae under Cd exposure. They could also provide the basic materials for further mining of other functional genes associated with the response of the antioxidant defense to heavy metal stress.

## 2. Materials and Methods

### 2.1. Larval Rearing and Treatment with Cd

The clams (*Meretrix meretrix*) taken around the period of spawning were incubated in a cement tank (8 × 3 × 1.5 m) at 28 ± 1 °C in the shellfish breeding base of Qingjiang at the Zhejiang Mariculture Research Institute. Sexually matured clams were removed from the tank, placed in the shade with a net for 5–6 h, and then stimulated by flowing seawater to induce spawning. After approximately 2–3 h, 12 clams were randomly selected and individually placed in 1-L beakers, each containing 400 mL of seawater, and the sex of each clam was determined by visual inspection of the gametes it produced. All experimental procedures were approved by Wenzhou University’s Animal Ethical and Welfare Committee (No. WZU-2023-103).

Sperms and eggs were collected from three male clams and three female clams, respectively, and then placed in a 100-L plastic tank containing 50-L filtered seawater without or with 27.2 μg L^−1^ CdCl_2_. The concentration of Cd^2+^ was based on 2/5 of the LC_50_ (68 μg L^−1^) for 96 h of exposure obtained for *M. meretrix* larvae [34]. This experiment had three replicates. About 1 g of fertilized eggs from each plastic tank was collected with a 300-mesh net and sampled using 5 pipettes one hour after the mixing of sperm and eggs in water without or with Cd. D-shaped larvae, pediveligers, and postlarvae were collected at 24, 96, and 168 h after the mixing of sperm and eggs. These major larval stages of *M. meretrix* development were chosen because the changes in morphology could be clearly identified under a microscope. All specimens were transferred to cryovials and frozen in liquid nitrogen. The larvae were raised at 28 ± 1 °C, 20‰ salinity, and pH 7.8, and the culture medium was replaced daily with fresh seawater containing no Cd^2+^ or 27.2 μg L^−1^ Cd^2+^. The larvae were fed with *Isochrysis* spp. three times a day, beginning at the D-shaped larval stage. All collected fertilized egg and larval samples were divided into two parts: one part was stored in an ultra-low-temperature refrigerator for immunohistochemistry and molecular experiments, whereas the other part was transported to Novogene Bioinformatics Technology Co., Ltd. (Beijing, China) on dry ice for transcriptome sequencing. 

### 2.2. RNA Isolation, Library Construction, and Illumina Sequencing

Total RNA extraction, mRNA purification, cDNA library construction, and illumine sequencing were conducted by Novogene Bioinformatics Technology Co., Ltd. (Beijing, China). Briefly, the total RNA was extracted from the *M. meretrix* fertilized eggs and larvae using Trizol reagent (Invitrogen, Waltham, MA, USA). The concentration and integrity of the total RNA were assessed using the Agilent 2100 bioanalyzer (Agilent Technologies, Santa Clara, CA, USA). Furthermore, the mRNA with a poly-A tail was enriched by poly-T oligo-attached magnetic beads. The purified mRNA was fragmented with divalent cations in the NEB Fragmentation Buffer, and the mRNA-seq libraries were constructed using the Library Prep Kit (NEB, Ipswich, MA, USA). Qubit 2.0 Fluorometer (Life Technologies, Carlsbad, CA, USA) and an Agilent Bioanalyzer 2100 system were used to qualify and quantify the sample libraries. The obtained libraries were sequenced and analyzed by an Illumina NovaSeq 6000 (Illumina, San Diego, CA, USA). 

### 2.3. Transcriptomic Data Quality Control and Assembly

For transcriptome analysis without a reference genome, the sequences obtained were spliced into transcripts, and the transcripts were subjected to hierarchical clustering and aggregated sequence analysis using the Trinity software package (version 2.4.0). After raw data filtering, checking for sequencing error rate and GC content distribution, and removing reads containing adapters and low-quality reads (including reads with a ratio of N greater than 10% and Q (quality value) ≤ 10 whose bases account for more than 50% of the entire read), transcriptome de novo assembly was accomplished using Trinity [35] with min_kmer_cov set to 2 by default and all other parameters set to default. Trinity combines three independent software modules: Inchworm, Chrysalis, and Butterfy, applied sequentially to process large volumes of RNA-seq reads. Then the transcripts were further aggregated into unigenes by Corset software (version 4.6) [36]. 

### 2.4. Gene Function Annotation

Unigenes were blasted against software (version V2.2.28+, comparison standard: E-value ≤ 1 × 10^−5^) and annotated based on seven databases, including NCBI non-redundant protein sequences (Nr), NCBI nucleotide sequences (Nt), Protein Family Database (Pfam, http://pfam.sanger.ac.uk/ accessed on 7 November 2022), EuKaryotic Ortholog Groups/Clusters of Orthologous Groups of proteins (KOG/COG, http://www.ncbi.nlm.nih.gov/CO/ accessed on 7 July 2021), Swiss-Prot (http://www.ebi.ac.uk/uniprot/ accessed on 7 December 2022), Kyoto Encyclopedia of Genes and Genomes (KEGG, http://www.genome.jp/kegg/ accessed on 17 December 2021), and Gene Ontology (GO, http://www.geneontology.org/ accessed on 7 November 2022). In addition, the clean reads were mapped to the reference unigene set, and the read number of gene expression in each sample was counted by the RSEM software (version 1.2.15) [37]. GO and KEGG pathway enrichment of DEGs were analyzed by GOseq (version 1.10.0), topGO (version 2.10.0), and KOBAS software (version 2.0.12), respectively.

### 2.5. Enrichment Analysis and Functional Annotation of Differentially Expressed Genes 

The gene expression profiles of *M. meretrix* fertilized eggs and larvae at different stages of development were analyzed, and the differentially expressed genes (DEGs) were selected and annotated. Then, through Gene Ontology (GO) analysis, the enriched GO classification items of the DEGs were identified and their functions determined. Based on the analyses performed with the Kyoto Encyclopedia of Genes and Genomes (KEGG), the pathways of the enriched DEGs were determined, and the genes that are involved in different steps of the metabolic pathways were investigated to correlate a particular gene or set of genes with a particular biological phenomenon.

### 2.6. Validation of Differentially Expressed Genes by qRT-PCR 

To verify the quality and accuracy of the transcriptome sequencing results obtained, six random DEGs and two specific genes related to oxidative stress were selected for a quantitative real-time PCR (*q*RT-PCR) assay. These selected genes were based on gene functional annotation and differential expression analysis. The samples used for validation were the remaining samples from the same batch used for transcriptome sequencing. 

The primers were designed by Primer Premier 5 software, and β-actin was used as the internal reference gene. The designed primers were synthesized by Sangon Biotech Co., Ltd., Shanghai, China, and are shown in Table 1. The amplification reaction for *q*RT-PCR (Roche, LC480) was carried out as follows: pre-denaturation at 95 °C for 5 min, followed by 40 cycles of denaturation at 95 °C for 15 s, annealing at 55 °C for 15 s, and extension at 72 °C for 20 s. A negative control was included, in which the sample contained no cDNA. The relative expression level of a gene was calculated by the 2^−∆∆Ct^ method. 

### 2.7. Immunohistochemical Detection of Nrf2 in Different Stages of M. meretrix Larvae

After collection, the fertilized eggs were directly fixed in a 4% paraformaldehyde-PBS solution overnight. D-shaped larvae, pediveligers, and postlarvae from the treatment and control groups were collected into beakers and were anesthetized by adding drops of 1 mol L^−1^ MgCl_2_ to a final concentration of 7% in seawater to induce an opening of the shells, and then were immediately followed by fixation with a 4% paraformaldehyde-PBS solution overnight at 4 °C. Both the fixed fertilized eggs and larvae were then transferred to 100% methanol and stored in a −20 °C refrigerator until use. Immunohistochemical analysis was carried out according to the method described by Wang [34]. Briefly, the fixed fertilized eggs and larvae were gradually rehydrated with a graded methanol-PBS solution (methanol concentration: 95%, 90%, 75%, 50%, 25%) and washed three times with 0.01 M PBS (pH 7.4). After incubation in 3% H_2_O_2_ for 25 min, the larvae were decalcified with a 10% EDTA-PBS solution for 30–45 min and then incubated with TBS-10% goat serum for 1 h to block any non-specific reaction. The fertilized eggs and larvae were then incubated with the Nrf2 IgG antibody (1:200 dilution) at 37 °C for 4 h. Immediately, the samples were washed three times with PBS and incubated with HRP-labeled goat anti-rabbit IgG (1:200 dilution) at room temperature for 3 h. After that, the samples were washed three times with PBS, added to the chromogenic substrate solution, and stained for about an hour. The samples were then washed three times with PBS and then transferred to a glycerol-PBS (3:1) solution for microscopic observation and image recording.

### 2.8. Data Processing

The SPSS 25 software package was used to analyze experimental data. Differences among groups were determined by one-way ANOVA and then by post hoc analysis using the least significant difference (LSD) test. Relevant graphs were drawn with Origin 2021. Data were expressed as mean ± SD, and statistical significance was considered at the *p* < 0.05 level. 

## 3. Results

### 3.1. Summary of Data Quality

The quality of sample sequencing data is shown in Table 2. The raw reads generated from the transcriptome of each sample ranged from 21,201,221 to 23,160,593. The clean reads ranged from 20,766,808 to 22,752,311, whereas the clean bases ranged from 6.23 G to 6.83 G, and the error rate of sequencing was 0.3%. The proportions of Q20 and Q30 exceeded 96.62% and 90.73%, respectively, and the GC percentages of the clean reads were from 35.07% to 39.58%. According to these data, the results of the transcriptome sequencing could be considered reliable.

### 3.2. Analysis of Differentially Expressed Genes 

#### 3.2.1. Statistics of Differential Expressed Genes 

Overall, 1915, 9196, 1620, and 19,183 genes were differently expressed in the fertilized eggs, D-shaped larvae, pediveligers, and postlarvae, respectively, after treatment with Cd^2+^ (Figure 1). Among the differentially expressed genes (DEGs) in the fertilized eggs, 209 were up-regulated and 1706 were down-regulated genes (Figure 1A). In the case of the D-shaped larvae, 3544 DEGs were up-regulated and 5652 were down-regulated genes (Figure 1B). As for the pediveligers, 1343 DEGs were up-regulated and 277 were down-regulated genes (Figure 1C), and finally, for the postlarvae, 9467 DEGs were up-regulated and 9716 were down-regulated genes (Figure 1D). It is obvious from the data that there was no particular pattern in the changes in gene expression in the various stages of development upon exposure to Cd, with the least change occurring in the pediveliger stage and the greatest change in the postlarva stage.

#### 3.2.2. GO and KEGG Functional Classification

In order to further determine what possible biological changes might be associated with the DEGs detected for each of the development stages of *M. meretrix*, spliced unigenes were compared and analyzed with the databases of GO and KEGG. 

In GO analysis, unigenes were assigned to the biological process (BP), cell component (CC), and molecular function (MF) (Figure 2). In the fertilized egg, pediveliger, and postlarva, DEGs that were associated with BP made up the largest number, and the majority of these were clustered in the metabolic and biosynthetic processes (Figure 2A,C,D). As for the D-shaped larva, the DEGs that were associated with BP were clustered in the proteolysis and oxidation-reduction processes (Figure 2B). In the CC category, the majority of the DEGs detected in the fertilized eggs, pediveligers, and postlarvae were clustered in the intracellular and organelle components, while none of the DEGs detected in the D-shaped larvae were involved in these components. As for the MF category, the majority of DEGs were clustered as follows: peptidase activity and transcription regulator activity groups in the case of the fertilized egg; catalytic activity group for the D-shaped larva and pediveliger; and binding activity for the postlarva. More down-regulated genes than up-regulated genes were observed for the fertilized egg, D-shaped larva, and postlarva stages (Figure 2A,B,D), while the opposite trend was observed for the pediveliger (Figure 2C).

In the KEGG metabolic pathway analysis (Figure 3), there were 485, 1454, 278, and 1806 differential genes in the fertilized egg, D-shaped larva, pediveliger, and postlarva, respectively, and these all featured in the top 20 significant pathways. The DEGs detected in the fertilized egg (Figure 3A) and D-shaped larva stages (Figure 3B) were mainly distributed in the cytoskeleton-related pathways and oxidative phosphorylation-related pathways. On the other hand, the DEGs found in the pediveliger stage (Figure 3C) were mainly associated with the ribosome and amino acid degradation-related pathways, while the DEGs found in the postlarva (Figure 3D) were mainly associated with the endocytosis, lysosome, and MAPK signaling pathways.

#### 3.2.3. Commonly Regulated Differential Genes

The Venn diagram (Figure 4) shows the number of common genes that were differentially expressed at the different developmental stages of *M. meretrix* following Cd treatment. In total, 10 genes were found to be differentially expressed in all developmental stages. The locations and identities of these 10 DEGs are presented in Table 3. Although the analysis listed ten genes, these could be reduced to six genes based on their functions, and these included the genes for cytochrome c oxidase (CCO) subunits I, II, and III, NADH dehydrogenase (Ndh) subunits 4 and 5, complement component 3 precursor (pro-C3), alpha-2-macroglobulin (A2M), hemopexin (HPX), and serotransferrin (STF). All these genes are related to cellular respiration, ion transport, oxidative stress, and immunity. 

### 3.3. Quantitative RT-PCR Validation of the Differentially Expressed Genes 

The results obtained for the validation of the differential genes by qRT-PCR are shown in Figure 5, and the expression trends of genes verified by qRT-PCR were basically consistent with those in RNA-seq, indicating that the sequencing results of the transcriptome of the clam were accurate and reliable.

In the cases of *CCO* and *Ndh* genes, the transcript levels under no treatment exhibited a significant decrease at the D-shaped larva stage, followed by a rapid and significant increase to a level similar to that of the fertilized egg at the postlarva stage (Figure 5A,B). After treatment with Cd, the transcript levels of these two genes displayed as much as a 500-fold reduction, followed by a similar increase in magnitude at the D-shaped larva stage, and then decreased, eventually reaching a level somewhat lower than that found at the fertilized egg stage. In both genes, differences in transcript levels between untreated and Cd-treated were rather huge, suggesting that the expression of these two genes might be particularly susceptible to Cd-induction. 

The transcript levels of the *pro-C3*, *HPX*, *STF*, and *A2M* genes exhibited similar trends under no treatment. For each gene, the highest level was found in the D-shaped larvae and the lowest level in the pediveligers, and the expression levels were significantly different among the four different developmental stages of *M. meretrix* except for the expression level of pro-C3 in the fertilized eggs and D-shaped larvae (Figure 5C–F). Under Cd treatment, the transcript levels of *pro-C3*, *HPX*, *STF*, and *A2M* decreased with the onset of development, with the highest level in the fertilized eggs and the lowest in the postlarvae. Furthermore, the transcript levels of all four genes also rose significantly relative to no treatment in the case of fertilized eggs and pediveligers. However, the extent of the increase in *A2M* expression level in the pediveliger was so much higher, about 23 folds (Figure 5F), and this constituted the most noticeable difference from the trends observed for *pro-C3*, *HPX*, and *STF* between no treatment and Cd-treatment. Whereas in the case of postlarvae, the transcript levels of all four genes decreased significantly relative to no treatment, with the reduction being most pronounced for *ProC3*, with as much as eight folds (Figure 5C). As for D-shaped larvae, *HPX*, *STF*, and *A2M* transcript levels decreased significantly following Cd-treatment compared with no treatment, but the transcript of *pro-C3* remained somewhat similar between Cd-treatment and no treatment. 

According to the transcript profiles, the changes in gene expression levels for the six genes across different developmental stages did not appear to follow any particular pattern, with Cd exposure resulting in increases for certain genes while decreases for others, and the extent of the increase or decrease was also different for different developmental stages, suggesting that Cd could at least impact the expression of the genes, consistent with the RNA-seq result.

### 3.4. Changes in MT and Nrf2 mRNA Levels 

Although RNA-seq showed that *MT* and *Nrf2* were not differentially expressed in all different stages of *M. meretrix* development, the expression of either gene was upregulated or downregulated when each of the larval stages was compared with the fertilized egg stage under Cd stress (Cluster-27498.34214, Cluster-27498.13667), suggesting *MT* and *Nrf2* could play an important role in *M. meretrix* larvae under Cd stress. Only slight variations in *MT* mRNA levels were detected across the four stages of *M. meretrix* development in the absence of Cd exposure through *q*RT-PCR, and the differences among the different stages were not significant (Figure 6A). However, after exposure to Cd, significant increases in *MT* mRNA levels occurred in the fertilized eggs, pediveligers, and postlarvae compared with no Cd exposure, with the postlarvae showing the greatest increase, reaching more than three-fold the level seen under no Cd treatment. Some increase also occurred in the D-shaped larvae, but the increase was not significant. Changes in *MT* mRNA across the different stages of *M. meretrix* development in response to Cd exposure were greater, with Cd having a stimulating effect, consistent with the induction of *MT* expression by Cd reported in our previous work [17]. 

In the control groups, pediveligers had the highest level of *Nrf2* mRNA, followed by postlarvae, while both fertilized eggs and D-shaped larvae had the least *Nrf2* mRNA (*p* < 0.05). Differences were detected between the pediveliger stage and each of the other stages. However, no significant difference was detected between the fertilized egg and D-shaped larva stages (white columns in Figure 6B). In the Cd-treated groups, there were significant (*p* < 0.05) differences in *Nrf2* mRNA levels among the different developmental stages, with the D-shaped larvae having the highest and that of the fertilized eggs having the lowest (black columns in Figure 6B). The level of *Nrf2* mRNA in the Cd-treated group was significantly lower than that in the control group at both the pediveliger and postlarva stages, but it was significantly higher than that of the control group at the D-shaped larva stage (Figure 6B).

### 3.5. Nrf2 Protein Immunohistochemistry in Different Developmental Stages of M. meretrix Larvae

In order to observe the stress effect of Cd on *M. meretrix*, the expression of Nrf2 protein in *M. meretrix* at different developmental stages was also analyzed by immunohistochemical assay. In this assay, when Nrf2 protein binds to the antibody, it shows a light brown color, and the darker the color, the more protein expression there is. Overall, the result showed that the protein level of Nrf2 increased during the development of *M. meretrix*, as shown by more intense brown dots as the larva developed in the absence of Cd treatment (Figure 7A,C,E,G). For the fertilized egg and D-shaped larva stages, no significant difference in the level of Nrf2 protein was detected between the absence and presence of Cd^2+^ treatment, as seen by the lack of difference in the intensity of the brown dots between the two (Figure 7). As for the pediveliger and postlarva stages, the intensity of brown dots increased in the presence of Cd^2+^ treatment, suggesting a significant increase in the level of Nrf2 protein in the pediveliger and postlarva stages when subjected to Cd treatment. The result appeared to suggest that Nrf2 protein expression increased during the development of *M. meretrix* and that Cd could induce an increase in the expression of Nrf2 protein in the larvae, with the later stages of the development showing a more intense response.

## 4. Discussion

At present, the extensive use of Cd has posed a great threat to the environmental ecosystem and human health [2,5,6]. In some industrialized areas in China, the levels of Cd exceed more than 20-fold the range of the sea water according to the official standard (10 to 500 ng L^−1^) [32,38]. Several studies have shown that Cd generates some adverse effects on *M. meretrix*, and the acute toxicity varies in different tissues [8,9,10,17,29]. Cd toxicity has caused huge economic losses to molluscan cultivation [1,2]. Therefore, it is essential to explore the mechanisms associated with the response of mollusk larvae to metal stress. Such a response is usually not confined to a change in a single gene or signal pathway but rather to a change in the entire regulatory network triggered by changes in the external environment. Thus, RNA-seq is a useful technique to explore the changes associated with the response of an organism to environmental stresses, including metal, and the implicated molecular mechanisms [12,27,31]. 

In this study, transcriptome libraries were constructed for the fertilized egg and different larval stages of *M. meretrix* under Cd stress, and their responses to the Cd stress were analyzed based on the changes in the total transcript levels. A total of 31,914 genes were found to be differentially expressed when comparing Cd-stressed larvae with non-stressed larvae. Moreover, GO and KEGG signaling pathway analysis (Figure 2 and Figure 3) revealed different response mechanisms and signaling pathways for the *M. meretrix* larvae when subjected to Cd stress, depending on the developmental stages of the larvae. Many of the DEGs were found to be involved in cellular respiration, ion transport, oxidative stress, and the immune response. A previous transcriptomic analysis of *M. meretrix* larvae revealed a dramatic discrepancy in certain transcripts among the different larval stages, and these transcripts are related to different processes, such as development, growth, shell formation, and immune responses [26], with wide variation in the expression patterns of these genes among the different larval stages. Some genes (e.g., the insulin gene) are expressed in the early larval stage (pre-pediveliger stage), while others (e.g., calmodulin and lysozyme genes) are expressed in the late larval stage (post-pediveliger stage). The transcriptomic analysis of different development stages of the mussel *Mytilus galloprovincialis* also revealed very different profiles of gene expression among the different developmental stages [39]. Genes that are linked to processes such as differentiation and biosynthesis are highly expressed, while genes involved in processes related to immune responses are strongly downregulated. However, in *M. galloprovincialis*, immune competency is strongly linked to metamorphosis when the larva starts to exhibit increased expression of immune-related genes and respond to environmental signals such as the presence of pathogenic bacteria [40]. It appears that during the course of normal development, certain sets of genes might be preferentially expressed to aid in the processes of development and growth. However, in the presence of a stressor, the expression profile could change, and these changes would reflect the kind of responses that are induced by the stressor. 

Among the DEGs that were common to the four developmental stages of *M. meretrix*, the *CCO* and *Ndh* genes are important components of the respiratory chain, which plays an important role in biological respiration, providing energy for life activities, and maintaining the normal physiological activities of organisms [41,42]. CCO is the terminal complex of the respiratory chains in the mitochondria of nearly all eukaryotes. It catalyzes the reduction of molecular O_2_ to water using electrons from the respiratory chain, delivered via cytochrome C on the external surface of the inner mitochondrial membrane [43]. CCO plays a significant role in the mitochondrial respiratory chain and the inhibition of ROS production [9,44]. As a flavoprotein, Ndh contains iron-sulfur centers and is involved in the production of ROS in mitochondria, suggesting that Ndh may be related to oxidative stress [45]. In the control group, the expression levels of *CCO* and *Ndh* decreased at first and then increased during the development of *M. meretrix* (Figure 5A,B). These trends may be due to the need for the fertilized egg to undergo rapid respiration to meet rapid cell division and growth, and such processes could be facilitated by increased expression of the corresponding genes. With the development of *M. meretrix*, the rates of metabolic changes may slow down, leading to somewhat lower energy expenditure and thereby putting less demand on the activity of respiration-related genes. This may explain the downward trench of *CCO* and *Ndh* expression in D-shaped and pediveliger larvae. When the larvae developed into the postlarvae, they entered yet another rapid developmental stage, and at this stage, the respiratory system would be well-formed. This, together with the rapid development of the postlarvae into the adult form, may present the organism with yet another high energy demand. Hence, increased *CCO* and *Ndh* expression at this stage may seem logical. Yang et al. [46] also found that the *CCO* mRNA levels correspond to subunits I, II, and III were gradually increased in the early development of *R. venosa* and suggested this may be related to continuous energy conversion during development. Under Cd stress, the levels of *CCO* and *Ndh* mRNA decreased but increased in the D-shaped larva and pediveliger compared with no Cd stress. The stress exerted by Cd on the fertilized egg might have diverted its energy production ability to defense-related processes, thereby accounting for the greatly reduced *Ndh* expression. Moreover, the high level of *Ndh* expression in the absence of Cd stress could mean that the fertilized egg already contained a sufficient level of *Ndh* gene product (mRNA or protein) to cope with the effect of Cd stress, relieving the cell of the need to increase its *Ndh* expression. The increased *Ndh* expression levels in the D-shaped larva and pediveliger stages following Cd induction may suggest a greater mobilization of energy (ATP) to deal with the stress. With prolonged exposure to Cd, this led to increased Cd enrichment within the postlarvae, which may overwhelm the ability of the postlarave to cope with the metal stress, resulting in decreased expression of respiration-related genes. Therefore, the alteration of CCO and Ndh levels reflected the oxidative stress induced by Cd in clam larvae. Xu et al. [31] also found that exposure of the mussel *M. galloprovincialis* to Cd in the early life stages resulted in a change in NADH dehydrogenase expression, consistent with our observation.

Huan et al. [26] found that stress- and immune-related genes play important roles in the early development of *M. meretrix*. Similarly, we have also found variation in the expression of a few immune genes (*pro-C3*, *A2M*, *HPX*, and *STF*) in the different developmental stages of *M. meretrix* and how this expression was affected by Cd exposure. The four immune-related genes have different functions. The *pro-C3* gene codes for the complement factor 3 (C3) precursor. C3 is the most abundant complement protein in the blood and plays a central role in complement activation [47]. Pierron et al. [3] also suggested that C3 is an effective parameter for evaluating the potentially toxic effects of heavy metals on teleost fish immune responses. A2M, on the other hand, is the critical pan-protease inhibitor of the innate immune system [48]. A2M can bind to metal ions, in particular Cd^2+^, but it can also bind to Zn^2+^, Ni^2+^, and Pb^2+^ [49]. HPX is an important acute phase response protein that has multiple functions such as anti-apoptosis, participation in inflammation regulation, neuroprotection, regulation of intracellular signal transduction, and immune regulation [50,51]. HPX can bind strongly to Zn^2+^ and Ni^2+^ columns, but it somewhat binds weakly to Cd^2+^ [49]. STF is an important iron-metabolism protein and a known antimicrobial protein capable of preventing infection from several pathogens [52]. It plays a crucial role in contributing to the immune response and the host defense [52,53]. In the early stages of larval development, the disease resistance of *M. meretrix* is relatively weak, especially in the D-shaped larva stage, making it more sensitive to the environment. In the mussel *M. galloprovincialis*, the adult population has been shown to be closely related to the fluctuation in the number of D-shaped larvae [30], suggesting that the D-shaped larva constitutes a rather sensitive stage of development. Thus, changes in the levels of *pro-C3*, *HPX*, *STF*, and *A2M* mRNA were most sensitive in the D-shaped larvae of *M. meretrix* (Figure 5C–F), which displayed the highest increases in these gene transcripts, probably to allow the larvae to cope with the external environment and better protect themselves. Wu et al. [30] suggested that D-shaped larvae of *M. galloprovincialis* faced more pathogens since they started to take in foods from the environment, which then required higher activity of immune molecules. This may indicate that the formation of corresponding tissues and organs started in D-shaped larvae. Moreover, some key molecules in the development of the nervous system and some mesodermal organs started to express in D-shaped larvae and were up- or down-regulated in the pediveliger and postlarva stages [26]. Hence, the up- or down-regulation of these genes in the pediveliger and postlarva of *M. meretrix* may also be related to the growth and proliferation of some particular cells in these two developmental stages.

Under Cd treatment, the levels of *pro-C3*, *A2M*, *HPX*, and *STF* mRNA decreased as the larva developed, with the highest levels in the fertilized egg and the lowest in the postlarva. The results showed that Cd exposure inhibited the immunomodulatory process of *M. meretrix* larvae during development, and with the enrichment of Cd, the degree of inhibition of immune-related genes was enhanced. These results also demonstrated the differentially responsive mechanisms to Cd exposure in the early life stages of *M*. *meretrix*. The significant reduction in *pro-C3*, *A2M*, *HPX*, and *STF* mRNA levels seen in the postlarva following Cd treatment (Figure 5C) may indicate an inhibitory effect caused by the excessive Cd resulting from its accumulation within the postlarva over time. Guo et al. [54] found that exposure of the catfish *Peltobargus fulvidraco* to increased Pb concentrations also led to a decreased level of *pro-C3* mRNA in the tissues, a phenomenon that the authors attributed to the Pb-induced immunosuppressive effect. Additionally, immune cell necrosis has also been demonstrated in the fish *Myoxocephalus scorpius* following exposure to high Pb concentrations [55]. The different responsive mechanisms of immune-related genes may also involve behavioral, morphological, physiological, and biochemical characteristics that could differ in different developmental stages of *M. meretrix*. Obviously, from the fertilized egg to the postlarva, different characteristics, such as sizes, organ systems, and the time taken for Cd to reach target sites, might trigger different responsive mechanisms to deal with Cd exposure. Xu et al. [31] also found that Cd stress can induce the disruption of immune regulatory function in the larvae of the mussel *M. galloprovincialis*.

Beside these genes, which were related to oxidative stress and immunity found in the DEGs in the four developmental stages of *M. meretrix*, two additional genes have been found in the clam larvae of the Cd^2+^-treated group: one codes for a metal-binding protein (metallothionein, MT), while the other codes for a transcription factor (Nrf2). Both MT and Nrf2 have previously been shown to be important for cytoprotection because of their roles in regulating the expression of antioxidant and detoxification genes in adult *M. meretrix* [8,17] and other aquatic animals [16,19]. The role of MT in alleviating oxidative damage induced by Cd has been demonstrated in *M. meretrix* and other bivalves via the simultaneous upregulation of MT expression and increased levels of oxidative damage, exemplified by increased levels of ROS and protein and lipid oxidation induced by Cd [8,10,25]. Furthermore, earlier studies that showed MT can function as a free radical scavenger in vitro provide further support for its role in alleviating oxidative damage [56]. MT and Nrf2 play a crucial role in the regulation of mitochondrial function and as master regulators of cellular redox homeostasis, as well as in the stimulation of the expression of an array of antioxidant response or immunity-related genes [10,15,20,24,57]. In this study, Cd stress also caused an up-regulated level of *MT* expression regardless of the stage of development, with the postlarva stage showing the greatest increase (Figure 6A), confirming its vital role in alleviating oxidative damage. MTs, as a superfamily of metal-binding proteins, have high heavy metal binding ability [21,22]. The induction of MT by Cd is a well-known response involved in protection against toxic heavy metals [8,25]. Huang et al. [23] found that the addition of exogenous MT can reduce the Cd content in the kidney and spleen of Cd-treated grass carp (*Ctenopharyngodon idellus*), alleviate their tissue damage, reduce the percentage of apoptosis, and restore the activity of immune-related enzymes. In the mussel *M. galloprovincialis*, *MT* also plays a role in the sequestration of reactive oxygen species (ROS), which are continuously produced in D-shape larvae exposed to Cd [31]. In mussels, an increased expression of *MT*, a defense mechanism against heavy metals and oxidative stress, suggests that activation of this defense mechanism would restrict the frequency range of ATP oscillations [58]. The significant up-regulation of *MT* confirmed the enhanced energy demand and mobilized detoxification mechanisms in *M. meretrix* larvae exposed to Cd. 

As an important regulatory factor, Nrf2 plays an important role in cellular antioxidation defense mechanisms and immune responses, such as heme, iron, and other cellular pro-inflammatory stress [11,19]. In this study, the expression of the *Nrf2* gene at the fertilized egg and D-shaped larval stages was significantly increased under Cd stress, while at the pediveliger and postlarval stages, such a phenomenon did not happen since the expression of the *Nrf2* gene at these stages was lower under Cd stress compared with no Cd stress (Figure 6B). Nrf2 signaling is critical to the defense against oxidative stress and can be initiated by many toxic substances, including pesticides and heavy metals. Down-regulation of the *Nrf2* gene represents a reduction in antioxidant capacity [24]. In the early development of *M. meretrix*, it is highly sensitive to cadmium, and, therefore, *Nrf2* expression is more likely to be induced than at the later stage of development, consistent with the immunohistochemical results observed for the pediveligers and postlarvae (Figure 7). A more developed immune system means that other ways to deal with these foreign harmful substances also become available, such as some low-molecular-weight compounds (vitamins A, C, E, and uric acid) synthesized to resist ROS produced by Cd stress [23,25].

Beside transcriptomic analysis, proteomic analysis has also been used to study changes in protein expression in bivalves subjected to environmental stresses, and this type of investigation has been recently discussed in a review by Balbi et al. [59]. Much of the work in this field appears to focus on the effects of pollutants, including pathogenic bacteria, on the immune response of bivalves. For example, the investigation into the effect of CO_2_-induced ocean acidification (OA) on the immune function of the Pacific oyster *Crasostrea gigas* against *Vibrio splendidus* revealed significant upregulation of superoxide dismutase (SOD) but a decreased abundance of catalase and GST, which indicates the antioxidant defense might have been overwhelmed by the effects of OA and the *V. splendidus* challenge [60]. Similarly, glutathione, catalase, and superoxide dismutase are also overexpressed in the mussel in response to contaminants ranging from heavy metals to organic xenobiotics [61]. As proteomic and transcriptomic analyses are both concerned with changes in gene expression in response to a stimulus, the nature of change is not the same since a change in protein level may not necessarily reflect a change in protein synthesis. Rather, it could well be due to a change in the stability of the protein, thereby resulting in a change in the abundance of the protein over time. This could be a possible reason for the high increase in Nrf2 protein in *M. meretrix* larvae without a corresponding increase in mRNA level in response to Cd exposure (Figure 6 and Figure 7). Perhaps a combined approach of transcriptomic and proteomic analysis would provide better insight into the extent of cellular functional changes when the organism is subjected to environmental stress.

## 5. Conclusions

In conclusion, we explored the effect of Cd on the development of *M. meretrix* larvae by using transcriptome sequencing technology. A total of 31,914 differential genes were obtained following the exposure of the different development stages of *M. meretrix* to Cd. These genes function in various biological processes, including structural reorganization, oxidative phosphorylation, cellular respiration, and oxidative stress. The oxidative stress-related genes of *M. meretrix* were found to be significantly affected. Our results indicated that multiple biological process adjustments may occur in the *M. merertrix* larvae after Cd stress. Although what we have detected were changes in the expression of genes predominantly related to oxidative stress management, the functions of these genes are not restricted to oxidative stress but may also be involved in immune response. Thus, toxic metal stress should be considered a multiple-stress-inducing factor, and more in-depth study is required to further clarify the molecular mechanism other than that related to oxidative stress.

## Figures and Tables

**Figure 1 animals-14-00352-f001:**
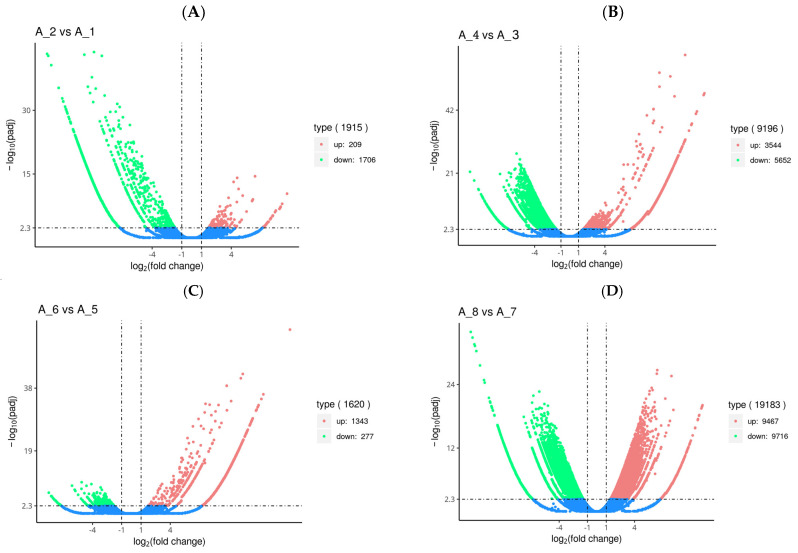
Volcano plot of differentially expressed genes (DEGs) in the fertilized egg (**A**), D-shaped larva (**B**), pediveliger (**C**), and postlarva (**D**) stages. Note: Log_2_ (fold change) (*x*-axis, representing the change of gene expression factor in different samples) versus log10 (padj-corrected *p*-value for multiple hypothesis testing) (*y*-axis, representing the significance level of expression difference). The red dot and the green dot represent significant up-regulated and down-regulated DEGs, respectively. The blue dot represents genes with no significant differences.

**Figure 2 animals-14-00352-f002:**
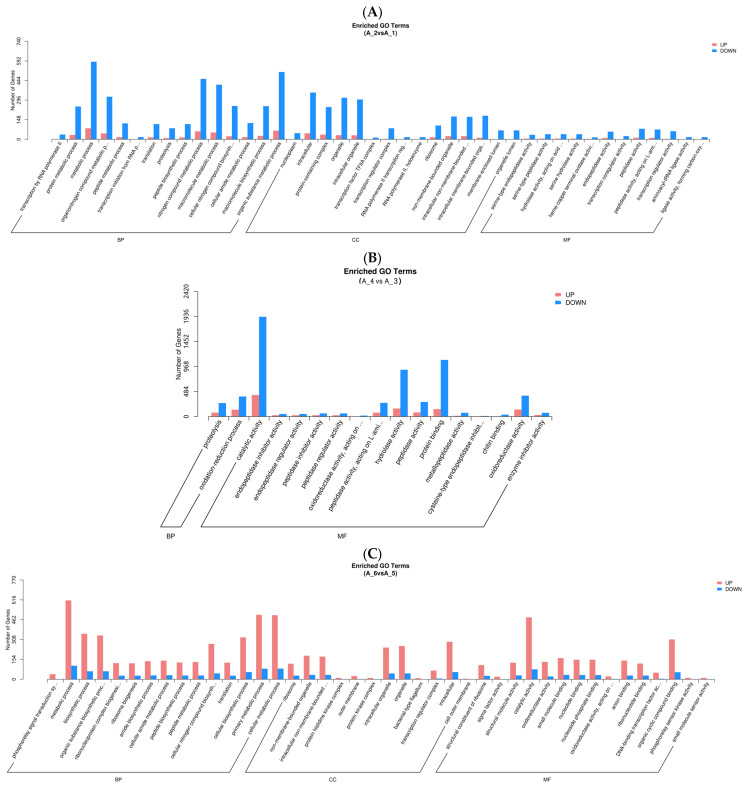
Functional enrichment analysis of GO differential genes. Note: The *x*-axis represents subcategories, and the *y*-axis represents the number of genes that GO term in a category. (**A**) fertilized egg; (**B**) D-shaped larva; (**C**) pediveliger; (**D**) postlarva.

**Figure 3 animals-14-00352-f003:**
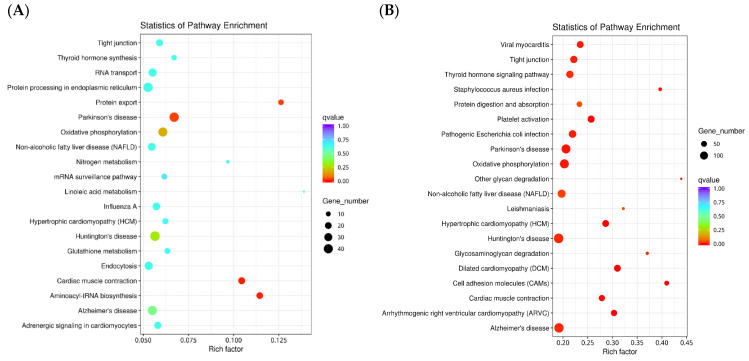
Scatter diagram of the KEGG pathway enrichment. Note: The *x*-axis represents the rich factor corresponding to the pathway, and the *y*-axis represents the pathway name. Q-values are represented by colors (blue: high; red: low), and the lower the q-value, the more significant the enrichment. Dot size represents the number of differential genes contained in each pathway, and the larger the dots, the more differential genes. (**A**) fertilized egg; (**B**) D-shaped larva; (**C**) pediveliger; (**D**) postlarva.

**Figure 4 animals-14-00352-f004:**
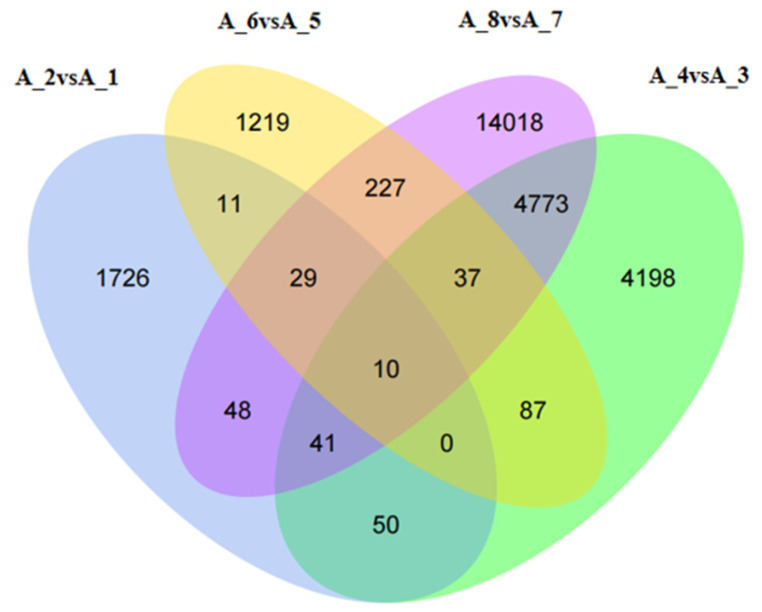
Venn diagrams of the differentially expressed genes. Note: A-1: Untreated fertilized egg; A-2: Cd-treated fertilized egg; A-3: Untreated D-shaped larva; A-4: Cd-treated D-shaped larva; A-5: Untreated pediveliger; A-6: Cd-treated pediveliger; A-7: Untreated postlarva; A-8: Cd-treated postlarva.

**Figure 5 animals-14-00352-f005:**
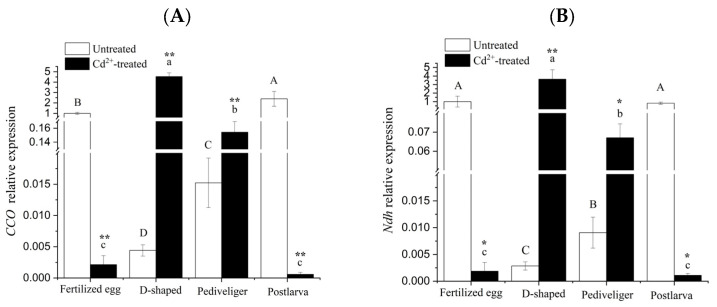
Quantitative RT-PCR validation of the differentially expressed genes. Note: (**A**) *CCO* mRNA; (**B**) *Ndh* mRNA; (**C**) *pro-C3* mRNA; (**D**) *HPX* mRNA; (**E**) *STF* mRNA; (**F**) *A2M* mRNA. Different letters indicate significant (*p* < 0.05) differences among groups. ‘*’ and ‘**’ indicate significant differences with respect to no Cd treatment for each development stage at the *p* < 0.05 and *p* < 0.01 levels, respectively.

**Figure 6 animals-14-00352-f006:**
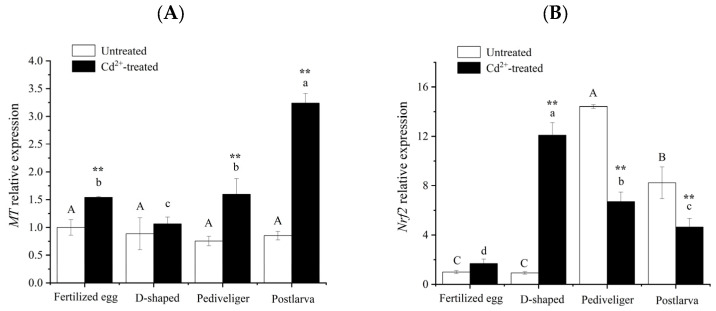
Changes in *MT* (**A**) and *Nrf2* (**B**) mRNA levels in *Meretrix meretrix.* at different stages of development. Note: Different letters indicate significant (*p* < 0.05) differences among groups. ‘**’ indicates significant differences with respect to no Cd treatment for each development stage at the *p* < 0.01 level.

**Figure 7 animals-14-00352-f007:**
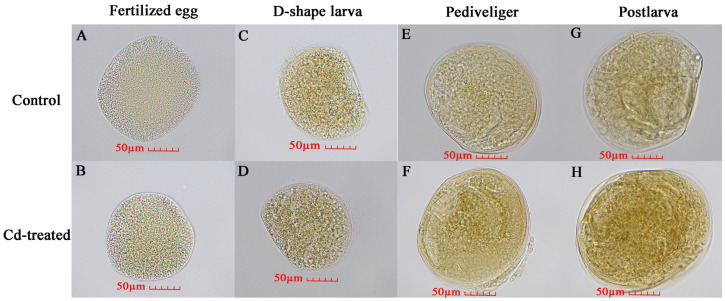
Nrf2 protein in different developmental stages of *Meretrix meretrix* as detected by immunochemical assay. Note: The brown color means the binding of Nrf2 protein with antibodies, and the darker the color, the higher the protein expression is. (**A**) Untreated fertilized egg; (**B**) Cd-treated fertilized egg; (**C**) Untreated D-shaped larva; (**D**) Cd-treated D-shaped larva; (**E**) Untreated pediveliger; (**F**) Cd-treated pediveliger; (**G**) Untreated postlarva; (**H**) Cd-treated postlarva.

**Table 1 animals-14-00352-t001:** Primer information.

Primer Information	Gene Symbol	Sequence (5′–3′)
NADH dehydrogenase-R	*Ndh*	GCAAACGGAGCCTCAATA
NADH dehydrogenase-F		AAGAATCGGGTCAAGGTG
Cytochrome c oxidase-R	*CCO*	AGGACAATGGGCATAAAG
Cytochrome c oxidase-F		GGGCACCAATGATACTGAA
Alpha-2-macroglobulin-R	*A2M*	GCTAAGACGACATAGGCACT
Alpha-2-macroglobulin-F		GAGAAACGAAATCCTGAAA
Hemopexin-R	*HPX*	GCAGTAGTAGCGTTCAAGC
Hemopexin-F		AATCCCATACCCACCAGA
Serotransferrin-R	*STF*	TTTCCAGTGCCTCGTTGA
Serotransferrin-F	*pro-C3*	AGGTGGCGAACTCGGTTA
Complement C3 precursor-R		ACAGAGTAGGGCAGTCGC
Complement C3 precursor-F		TCGTGAGCAGCACAGAAG
Mm β-actin-R		TTGTCTGGTGGTTCAACTATG
Mm β-actin-F		TCCACATCTGCTGGAAGGTG
Nrf2-R	*Nrf2*	TTTTACCCGCAGCAACTA
Nrf2-F		ATTCTCGTGCCTTCGTTT
MT-R	*MT*	GCAAACAACTTTACACCCTGGAC
MT-F		CGAGGACTGTTCATCAACCACTG

**Table 2 animals-14-00352-t002:** Summary of sample sequencing data quality.

Sample	Raw Reads	Clean Reads	Clean Bases	Error-Rate	Q20	Q30	GC-pct
A-1	22,908,233	22,608,398	6.78 G	0.03	96.62	90.84	35.66
A-2	22,807,303	22,438,671	6.73 G	0.03	96.76	91.07	36.09
A-3	21,689,381	21,415,748	6.42 G	0.03	97.29	92.28	39.58
A-4	22,456,576	22,190,209	6.66 G	0.03	96.62	90.73	35.07
A-5	23,112,929	22,704,192	6.81 G	0.03	97.31	92.30	35.96
A-6	22,529,731	22,138,576	6.64 G	0.03	96.88	91.42	35.47
A-7	21,201,221	20,766,808	6.23 G	0.03	96.70	91.09	36.61
A-8	23,160,593	22,752,311	6.83 G	0.03	97.15	92.04	39.18

Note: A-1: Untreated fertilized egg; A-2: Cd-treated fertilized egg; A-3: untreated D-shaped larva; A-4: Cd-treated D-shaped larva; A-5: untreated pediveliger; A-6: Cd-treated pediveliger; A-7: untreated postlarva; A-8: Cd-treated postlarva.

**Table 3 animals-14-00352-t003:** Differential gene information.

Gene ID	FPKM A-2 vs. A-1	FPKM A-4 vs. A-3	FPKM A-6 vs. A-5	FPKM A-8 vs. A-7	Description
Cluster-27498.58936	Down	Up	Up	Down	Cytochrome c oxidase subunit Ⅰ
Cluster-27498.35815	Down	Up	Up	Down	Cytochrome c oxidase subunit Ⅲ
Cluster-27498.57862	Down	Up	Up	Down	NADH dehydrogenase subunit 5
Cluster-27498.58765	Down	Up	Up	Down	Cytochrome c oxidase subunit Ⅱ
Cluster-27498.2982	Down	Up	Up	Down	NADH dehydrogenase subunit 4
Cluster-27498.46352	Up	Up	Up	Down	Cytochrome c oxidase subunit Ⅲ
Cluster-52252.0	Up	Down	Up	Down	Complement C3 precursor
Cluster-27498.52394	Up	Down	Up	Down	alpha-2-macroglobulin
Cluster-51975.0	Up	Down	Up	Down	Hemopexin
Cluster-27498.31390	Up	Down	Up	Down	Serotransferrin

Note: A-1: Untreated fertilized egg; A-2: Cd-treated fertilized egg; A-3: Untreated D-shaped larva; A-4: Cd-treated D-shaped larva; A-5: Untreated pediveliger; A-6: Cd-treated pediveliger; A-7: Untreated postlarva; A-8: Cd-treated postlarva.

## Data Availability

All authors guarantee that all data and materials support our published claims, and the data are available by contacting XP Ying (xpying2008@wzu.edu.cn).

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
