# Peer review of "Transcriptome Analysis and Identification of Cadmium-Induced Oxidative Stress Response Genes in Different Meretrix meretrix Developmental Stages"

_animals, 2024, doi:10.3390/ani14020352_

Round 1

Reviewer 1 Report

Comments and Suggestions for Authors

Authors reported the transcriptome data on the effect of Cd on each developmental stage of the industrially important clam, M. meretrix. Cd is known to accumulate in shellfish, and environmental levels of Cd may also be environmentally relevant in China. The manuscript looks good, but there are some problems that need to be corrected. The quality of the figures should also be improved.

General points:

1)    It seems that some proteome studies have been carried out in the clam including Meretrix meretrix. Authors should compare the present results with these studies, if necessary.

2)    2) Authors mentioned that to elucidate the underlying molecular mechanisms of oxidative stress in the developmental processes of the clam exposed to Cd in Abstract. However, this is different from the sentences in the last paragraph in Introduction and Conclusion of this manuscript. First of all, it is difficult to explain all the responses of the clam to Cd2 by oxidative stress alone. The authors should use antioxidants to confirm the involvement of oxidative stress.

Specific points:

1)    A sentence in L15: Commercially important should be mentioned strictly. Please add “aquaculture” in this sentence.

2)    A sentence from L15: “To elucidate the underlying molecular mechanisms of oxidative stress” should be modified like “To elucidate the underlying molecular mechanisms of Cd2+”. Although oxidative stress may be a major factor, the fact that Cd exhibits other effects is also noted in this Abstract (L32-35).

3)    L50: A phrase “sequestering metal ions” means the involvement of metallothionein?

4)    L101: Please insert a space between the number and "m".

5)    L157: Please insert a space between “In” and “addition”.

6)    L173: An extra space before “oxidative”?

7)    2.8. Data processing from L193: Which post-hoc test did Authors use with ANOVA?

8)    L213: Why did the post larval stage show an enormous number of genes that were differentially expressed compared to the other stages? Is this common for other xenobiotics?

9)    Figure 7: It is difficult to confirm positive signals for Nrf2. Can the authors provide other images or immunoblot images? First of all, these images are of low quality and I cannot read some words (in red) at the bottom of each panel. If these are not needed, please remove them. In the legend, do the Cd levels (10 to 500 ng L −1) mean those in the river and lake?

10)  A sentence from L462: Did the authors check for clam death? An overloaded CD could kill the clam.

11)  A sentence from 485: The reference is needed.

12)  A sentence from L533: Authors should provide any evidence that MTs have some role in oxidative damage during Cd exposure. Some references or Authors’ experimental evidence are enough. Which evidence can Authors present in the linkage of induction of MTs and the antioxidant effects?  These may have occurred in parallel.

Author Response

Responds to the reviewer’s comments:

Reviewer 1:

Comments and Suggestions for Authors

Authors reported the transcriptome data on the effect of Cd on each developmental stage of the industrially important clam, M. meretrix. Cd is known to accumulate in shellfish, and environmental levels of Cd may also be environmentally relevant in China. The manuscript looks good, but there are some problems that need to be corrected. The quality of the figures should also be improved.

General points:

  • It seems that some proteome studies have been carried out in the clam including Meretrix meretrix. Authors should compare the present results with these studies, if necessary.

Response: Thanks for your valuable suggestion. We have added additional information on the some comparing with the proteome studies of clams in the revised manuscript (Page 15, Line 457-474).

2) Authors mentioned that to elucidate the underlying molecular mechanisms of oxidative stress in the developmental processes of the clam exposed to Cd in Abstract. However, this is different from the sentences in the last paragraph in Introduction and Conclusion of this manuscript. First of all, it is difficult to explain all the responses of the clam to Cd2 by oxidative stress alone. The authors should use antioxidants to confirm the involvement of oxidative stress.

Response: Thank you for your helpful advice. We have modified the sentences in the last paragraph in Introduction (Page 3, Line 113-114) and Conclusion (Page 19, Line 649-650). Indeed, explaining the response of the clam to Cd toxicity based entirely on oxidative stress can be inadequate. However, oxidative stress does constitute a significant part of the response, making it a valid evaluation. Other potential responses may be inflammation based, which would seem to involve more immune-related genes. Although we have not directly measured antioxidant levels in this study, Cd is well known to induce an increase in antioxidant levels, such as SOD, CAT and GSH activities, which we and others have previously reported for M. meretrix in response to Cd exposure (Huang et al., 2020; Wang et al., 2010; Xia et al., 2016).

Huang, Y.; Tang, H.C.; Jin, J.Y.; Fan, M.B.; Chang, A.K.; Ying, X.P. Effects of waterborne cadmium exposure on its internal distribution in Meretrix meretrix and detoxification by metallothionein and antioxidant enzymes. Frontiers in Marine Science 2020, 7, 00502.

Wang, Q.; Wang, X.M.; Wang, X.Y.; Yang, H.S.; Liu, B.Z. Analysis of metallotionein expression and antioxidant enzyme activities in Meretrix meretrix larvae under sublethal cadmium exposure. Aquatic Toxicology 2010, 100, 321–328.

Xia, L.P.; Chen, S.H.; Dahms, H.U.; Ying, X.P.; Peng, X. Cadmium induced oxidative damage and apoptosis in the hepatopancreas of Meretrix meretrix. Ecotoxicology 2016, 5, 959–69.

Specific points:

  • A sentence in L15: Commercially important should be mentioned strictly. Please add “aquaculture” in this sentence.

Response: Thanks for your comments. We have added “aquaculture” in this sentence (Page 1, Line 29-30).

  • A sentence from L15: “To elucidate the underlying molecular mechanisms of oxidative stress” should be modified like “To elucidate the underlying molecular mechanisms of Cd2+”. Although oxidative stress may be a major factor, the fact that Cd exhibits other effects is also noted in this Abstract (L32-35).

Response: Thank you for this suggestion. We have modified the sentence as “To elucidate the underlying molecular mechanisms of Cd2+” (Page 1, Line 31).

  • L50: A phrase “sequestering metal ions” means the involvement of metallothionein?

Response: Yes, the “sequestering metal ions means the involvement of metallothionein”, and we have modified this sentence (Page 2, Line 68-70).

  • L101: Please insert a space between the number and "m".

Response: Thinks, we have inserted a space between the number and “m” (Page 3, Line 120).

  • L157: Please insert a space between “In” and “addition”.

Response: Thinks, we have inserted a space between “In” and “addition” (Page 3, Line 176).

  • L173: An extra space before “oxidative”?

Response: Thinks, we have deleted the space before “oxidative” (Page 4, Line 192).

  • 8. Data processing from L193: Which post-hoc test did Authors use with ANOVA?

Response: We have added the post-hoc test and modified the sentence of data processing (Page 6, Line 126-228).

  • L213: Why did the post larval stage show an enormous number of genes that were differentially expressed compared to the other stages? Is this common for other xenobiotics?

Response: Thank you for this query. We think the post larval stage is much more sophisticated in structure and functional competency compared with the other developmental stages, and therefore, a greater number of genes could be expressed to deal with external stimulus, which is Cd in this case (Page 15, Line 461-466). We are not sure about other xenobiotics, whether they also lead to a large number of genes being expressed in the post larval stage.

  • Figure 7: It is difficult to confirm positive signals for Nrf2. Can the authors provide other images or immunoblot images? First of all, these images are of low quality and I cannot read some words (in red) at the bottom of each panel. If these are not needed, please remove them. In the legend, do the Cd levels (10 to 500 ng L 1) mean those in the river and lake?

Response: Thank you for this suggestion, we have redone Fig. 7 to show more clearly the scale bars and larger text side in the revised manuscript (Page 14). The Cd levels (10 to 500 ng L 1) refer to the levels in the seawater chosen according references 32 and 38 (Page 15, Line 439 in the revised manuscript).

  • A sentence from L462: Did the authors check for clam death? An overloaded CD could kill the clam.

Response: Thanks for your comments. The concentration of Cd2+ in our study was 27.2 μg L 1 which corresponded to 2/5 of the LC50 (68 μg L 1) as determined for a period of 96 h. We did not observe any obvious death for the clams under this Cd concentration within the period of the experiment.

  • A sentence from 485: The reference is needed.

Response: Thank you for this suggestion, we have added one reference (Page 16, Line 530).

  • A sentence from L533: Authors should provide any evidence that MTs have some role in oxidative damage during Cd exposure. Some references or Authors’ experimental evidence are enough. Which evidence can Authors present in the linkage of induction of MTs and the antioxidant effects? These may have occurred in parallel.

Response: Thanks for your valuable suggestion. We have included information pertaining to the role of MT as an antioxidant in the revised manuscript (Page 17, Line 576-581).

Reviewer 2 Report

Comments and Suggestions for Authors

This paper deals with the responsive mechanisms to Cd stress in different Meretrix meretrix developmental stages. The levels of Cd in some industrialized areas in China exceed more than the official standard. Several studies have shown that the acute toxicity of Cd varies in different tissues. In this study, transcriptome libraries were constructed and used to screen for the stress related genes, and their potential roles were discussed. This could provide a step forward in elucidation of the molecular mechanism involved in immune response.

The description about all specimens used in the study seems to be too short and wrong. I assume the same number of eggs or larvae were used without or with Cd, but the number of specimens collected should have been different in both treatments. The authors called larvae collected 96 and 168 h after fertilization pediveligers and postlarvae, respectively. Pediveligers mean a stage of a veliger when it is able to crawl using its foot. I do not think 96 h larvae after fertilization have their foots. The swimming veligers are not postlarvae. The shell lengths of 24, 96 and 168 h larvae after fertilization are around 110 – 200 µm. The larvae shown in Fig.7 are too large. Shell lengths should be measured again.

In Fig.7, it is difficult to find where the positive signal is. I do not understand why Nrf2 protein is examined by immunochemical assay. In Methods, the procedure of IHC assay should be added. The authors claim that the brown dots (show them in Fig.7) were increased in Cd-treated groups for 96 and 168 h AF larvae. The level of Nrf2 mRNA in Cd-treated group was significantly lower than that in the control group at both 96 and 168 h larval stages in Fig.6. The difference between the gene and protein expressions should be discussed.

Lines419-422, I am puzzled as to why the authors referred Yang et al. [39].

Author Response

Reviewer 2:

Comments and Suggestions for Authors

This paper deals with the responsive mechanisms to Cd stress in different Meretrix meretrix developmental stages. The levels of Cd in some industrialized areas in China exceed more than the official standard. Several studies have shown that the acute toxicity of Cd varies in different tissues. In this study, transcriptome libraries were constructed and used to screen for the stress related genes, and their potential roles were discussed. This could provide a step forward in elucidation of the molecular mechanism involved in immune response.

1) The description about all specimens used in the study seems to be too short and wrong. I assume the same number of eggs or larvae were used without or with Cd, but the number of specimens collected should have been different in both treatments. The authors called larvae collected 96 and 168 h after fertilization pediveligers and postlarvae, respectively. Pediveligers mean a stage of a veliger when it is able to crawl using its foot. I do not think 96 h larvae after fertilization have their foots. The swimming veligers are not postlarvae. The shell lengths of 24, 96 and 168 h larvae after fertilization are around 110 – 200 µm. The larvae shown in Fig.7 are too large. Shell lengths should be measured again.

Response: Thank you for your comments. Regarding the description of the specimens, we have added a little bit more information, including specifying the number of specimens used in the experiment. As for the point you raised regarding the pediveliger that we used have no foot because 96 h post fertilization is too short, the foot primordium was actually present at this stage when examined by microscope event though the actual foot structure may not be obvious. Again, this has also been reported in the literature (Li et al., 2015). We named the specimens, which were based on the time after fertilization, in accordance with what other investigators have used in their papers (Huan et al., 2012; Wang et al., 2010). As for the shell lengths of Fig. 7 being too large, we are sorry to say this is a mistake resulting from the mislabeling. We have looked again and discovered that the scale bar in the previous version was incorrect, because it was about two-fold larger. We have now corrected this mistake (Page 14).

Huan, P.; Wang, H.X.; Liu, B.Z. Transcriptomic analysis of the clam Meretrix meretrix on different larval stages. Marine Biotechnology 2012, 14, 69–78.

Wang, Q.; Wang, X.M.; Wang, X.Y.; Yang, H.S.; Liu, B.Z. Analysis of metallotionein expression and antioxidant enzyme activities in Meretrix meretrix larvae under sublethal cadmium exposure. Aquatic Toxicology 2010, 100, 321–328.

Li, Z.M.; Qian, J.H.; Liu, Z.G.; Liu, J.S.; Li, Y.H. Development of embryo, larvae and spat of Meretrix lyrate (In Chinese). Marine Sciences 2015, 39 (7), 52–59

2) In Fig.7, it is difficult to find where the positive signal is. I do not understand why Nrf2 protein is examined by immunochemical assay. In Methods, the procedure of IHC assay should be added. The authors claim that the brown dots (show them in Fig.7) were increased in Cd-treated groups for 96 and 168 h AF larvae. The level of Nrf2 mRNA in Cd-treated group was significantly lower than that in the control group at both 96 and 168 h larval stages in Fig.6. The difference between the gene and protein expressions should be discussed.

Response: Thank you for this query. We have added the procedure of IHC assay (Page 5-6, Line 204-224) for Nrf2 to present extra data on the change of Nrf2 in response to Cd exposure in the different specimens (Page 14, Line 417-430). Surprisingly, the result from IHC assay does not seem to reflect that of qRT-PCR. At this stage our only explanation is the Cd exposure might somehow lead to the increased stability of Nrf2 protein despite its mRNA level being downregulated. We have inserted this information in the revised manuscript (Page 18, Line 616-635).

3) Lines419-422, I am puzzled as to why the authors referred Yang et al. [39].

Response: Thank you for your comments. We have cited other references in its place (Page 15, Line 457-474).

Round 2

Reviewer 1 Report

Comments and Suggestions for Authors

No comments.